# Lifestyle, Volition, and Well-Being Among Medical and Non-Medical University Students: A Preliminary Study

**DOI:** 10.3390/bs15111468

**Published:** 2025-10-28

**Authors:** Giovanna Zimatore, Ludovica Cardinali, Carlo Baldari, Manuela Minozzi, Valerio Bonavolontà, Piercesare Grimaldi, Laura Guidetti, Dafne Ferrari, Maria Chiara Gallotta

**Affiliations:** 1Department of Theoretical and Applied Sciences, DiSTA, eCampus University, 22060 Novedrate, Italy; giovanna.zimatore@uniecampus.it (G.Z.); carlo.baldari@uniecampus.it (C.B.); 2Department of Life Sciences, Health, and Health Professions, Link Campus University, 00165 Rome, Italy; l.cardinali@unilink.it (L.C.); m.minozzi@unilink.it (M.M.); p.grimaldi@unilink.it (P.G.); d.ferrari@unilink.it (D.F.); 3Department of Biotechnological and Applied Clinical Sciences, University of L’Aquila, 67100 L’Aquila, Italy; 4Department of Humanities, Movement and Education Sciences, University Niccolò Cusano, 00166 Rome, Italy; laura.guidetti@unicusano.it; 5Department of Physiology and Pharmacology “Vittorio Erspamer”, Sapienza University of Rome, 00185 Rome, Italy; mariachiara.gallotta@uniroma1.it

**Keywords:** physical activity, psychological well-being, university lifestyle, Mediterranean diet

## Abstract

Background: The transition to university represents a crucial period for the consolidation of health-related behaviors that may persist into adulthood. Examining gender and field-of-study differences can help identify factors shaping students’ well-being and lifestyle habits. Methods: This study assessed lifestyle behaviors and psychological well-being in 202 university students (125 females, 77 males; mean age = 19.76 ± 2.40) including 157 medical and 45 non-medical students. At the beginning and end of the academic year, participants completed questionnaires assessing diet (MEDAS), physical activity (IPAQ, VEQ-I), and psychological well-being (PSS-10, WHO-5, PGWB-S). Results: Males showed higher BMI, greater volitional resources supporting goal-directed behavior (VEQ-VF), and psychological well-being, while females reported greater stress (PSS-10) and lower volition (*p* < 0.05). Non-medical students scored higher on the WHO-5 (*p* = 0.041). Males and non-medical students were more frequently in the high IPAQ category (*p* < 0.01). Physical activity correlated positively with volition and well-being and negatively with stress. Conclusions: These results suggest that volitional resources support adaptive behaviors and are linked to both physical activity and psychological health. Promoting volitional skills, particularly among medical students and females, may enhance well-being and encourage healthier lifestyles during their university years.

## 1. Introduction

The transition to university life represents a key time when young adults develop habits that often last for years or decades ([31]). During this transitional phase, students gain autonomy while facing academic, social, and health-related challenges. As they adapt to increased responsibilities and reduced external support, research has shown the emergence of unhealthier behaviors including poorer diet, lower physical activity, insufficient sleep, and higher stress levels ([1], [2]).

Health sciences students present a particularly interesting case. While facing the same university challenges as their peers, these future healthcare providers study curricula focused specifically on health promotion and disease prevention. Students’ education in the health sciences promotes specialized knowledge and a professional identity oriented toward wellness advocacy ([17]; [28]; [34]). However, a critical question emerges regarding the extent to which this educational emphasis on health promotion is reflected in their personal behaviors and lifestyle choices ([15]). This inquiry holds significant implications for both educational practice and public health outcomes.

Even among health sciences students, challenges in consistently translating theoretical knowledge into personal health practices indicate that information provision alone may be insufficient to sustain behavioral change ([39]). This observation should not be interpreted as a limitation of current academic programs, which increasingly emphasize the integration of practical and experiential learning with theoretical content. Rather, it underscores the importance of adopting multidimensional educational strategies that also consider psychological, environmental, and social determinants of health behavior. Conversely, evidence of favorable health outcomes among these students could illuminate effective pedagogical components that might be adapted to broader university contexts, thereby informing institutional initiatives aimed at enhancing student well-being across disciplines.

This study comparatively examined the lifestyle behaviors and psychological well-being between medical sciences students and their peers in non-medical disciplines. Through analysis of Mediterranean diet adherence, physical activity engagement, stress levels, and subjective well-being, we investigated whether formal health education correlated with measurable differences in personal health practices and outcomes.

We hypothesize that medical sciences students may demonstrate enhanced health behaviors reflecting their specialized knowledge. Conversely, it may emerge that the mere possession of health-related knowledge, even when derived from advanced academic training, does not automatically translate into sustained behavioral change in the absence of supportive contextual and institutional frameworks. This perspective does not diminish the value of higher education but rather emphasizes the complexity of bridging the gap between knowledge and practice. Understanding this dynamic highlights the crucial role of universities in fostering not only cognitive competence, but also the environmental and psychosocial conditions that enable students to embody the health principles they are taught.

As a preliminary study, it is important to investigate the lifestyle habits of students enrolled in different faculties (medicine vs. others) at the beginning of their academic journey. In addition to disciplinary differences, examining gender-related variations provides a more comprehensive understanding of the health behaviors and well-being among university students. Recent research highlights that gender plays a significant role in shaping health awareness, engagement in physical activity, and psychological well-being. Female students often show greater attention to health and nutrition but also report higher levels of emotional distress and lower participation in physical activity compared with their male counterparts ([11]; [18]). Such findings suggest that gender influences not only individual behaviors but also the ways in which educational and psychosocial factors interact to shape students’ overall health and well-being ([41]).

The present study aimed to evaluate the lifestyle behaviors and psychological well-being among university students, with specific attention to the influence of physical activity and diet. Furthermore, it sought to explore differences between medical and non-medical students as well as gender-related variations to provide a more comprehensive understanding of how educational background and gender interact in shaping health-related behaviors and well-being.

## 2. Materials and Methods

In the present study, two groups were considered: a medical group including only students enrolled in the Medicine and Surgery program, excluding dentistry, and another group comprising students whose curricula encompassed elements of health education and disease prevention but did not lead to a qualification as a healthcare or clinical professional.

A total number of 202 (125 females, 77 males; age: 19.76 ± 2.40 years) participants were involved in this study. The volunteer participants were grouped into medical students (MED; *n* = 157, 102 females, 55 males; age 19.82 ± 2.67 years) and students from other academic programs (sport and exercises sciences, primary teacher education studies or others; all of these programs were named as Other; *n* = 45, 23 females, 22 males; age 19.56 ± 0.94 years).

At the beginning of the academic year (November 2024), six surveys were administered to evaluate the differences between groups. The tests were on eating habits, physical activities, and psychological variables (psychological well-being, perceived stress).

### 2.1. Anthropometric Measures

Regarding the subjects’ anthropometric measurements (i.e., weight and height, consequently body mass index (BMI)), were self-reported by the subjects. Therefore, if one of these two pieces of data was not indicated, the BMI could not be calculated. Moreover, gender was self-reported by the participants.

### 2.2. Mediterranean Diet Adherence

The Mediterranean Diet Adherence Screener (MEDAS) questionnaire is a tool used to assess an individual’s adherence to the Mediterranean diet ([27]). It consists of 14 questions related to dietary habits, assigning a score based on the frequency and quantity of consumption of specific Mediterranean diet foods. Each item refers to the habitual weekly consumption of key food groups and cooking habits characteristic of the Mediterranean diet such as olive oil, fruits, vegetables, fish, legumes, nuts, and moderate wine intake as well as the limited consumption of red meat and processed foods. It is widely used in epidemiological and clinical studies to assess dietary habits and the risk of cardiovascular, metabolic, and neurodegenerative diseases. It is also used by nutritionists and healthcare professionals to monitor dietary changes over time. For the interpretation of the score, we considered 1 point for each response if it met the recommendation, or 0 points if it does not meet the recommendation. The total score ranged from 0 to 14 points and was defined as 0–5 points for low adherence to the Mediterranean diet; 6–9 points for moderate adherence; ≥10 points for high adherence, associated with health benefits. The MEDAS has demonstrated high reliability (Cronbach’s α ≈ 0.7; ICC ≈ 0.8) ([30]) and good validity (*r* ≈ 0.5) ([27]) across Mediterranean and non-Mediterranean populations ([14]). In this study, participants were asked to report their usual dietary habits rather than short-term consumption (e.g., last 24–48 h), allowing the score to reflect general dietary patterns and long-term adherence rather than transient eating behaviors.

### 2.3. Physical Activity Levels (IPAQ)

Physical activity was self-reported using the International Physical Activity Questionnaire—Short Form (IPAQ-SF), covering the last 7 days’ activities in four domains: walking, moderate-intensity, vigorous-intensity, and time spent sitting ([24]). The IPAQ-SF is an internationally standardized instrument for assessing physical activity levels in adults and has demonstrated acceptable reliability and validity across various populations including university students ([5]; [24]). We employed the official IPAQ scoring protocol that determines the different metabolic equivalent of task (MET) for each domain or rather, walking was assigned 3.3 METs; moderate activity 4.0 METs; and vigorous activity 8.0 METs. Total MET-minutes/week was computed by summing (days × minutes per day × MET) for each intensity level; sitting time was reported as average daily minutes. Participants were categorized into physical activity levels according to the IPAQ scoring protocol. Based on their total MET-minutes/week, participants were classified into one of the following categories as follows. Low: not meeting the criteria for moderate or high activity; moderate: at least 600 MET-min/week, accumulated through ≥5 days of moderate activity or ≥3 days of vigorous activity or any combination; and high: ≥1500 MET-min/week of vigorous activity on ≥3 days or ≥3000 MET-min/week of any combination of activities ≥7 days.

The Italian version of the questionnaire has shown good psychometric properties, with test–retest reliability coefficients ranging between 0.73 and 0.90 and moderate correlations with objective measures of energy expenditure ([26]). For these reasons, the IPAQ-SF represents a widely used and reliable tool for evaluating habitual physical activity patterns in epidemiological and educational contexts.

### 2.4. VEQ-I

Volitional aspects related to physical activity were assessed using the Italian version of the Volition in Exercise Questionnaire (VEQ-I), which consists of eighteen items, each rated on a numeric scale from 0 (“does not match at all”) to 3 (“exactly matches”). The VEQ-I assesses six dimensions of volition including four volitional inhibition (VI) factors: Reasons, Postponing Training, Unrelated Thoughts, and Approval from Others, which hinder an individual’s ability to pursue exercise goals. It also includes two volitional facilitation (VF) factors: Self-Confidence and Coping with Failure, which support goal achievement and sustained engagement in physical activity ([13]).

The VEQ-I represents the validated Italian adaptation of the Volition in Exercise Questionnaire developed by [10] ([10]), designed to evaluate motivational and self-regulatory mechanisms underlying exercise behavior. The Italian version has shown good psychometric properties, with a six-factor structure confirmed through confirmatory factor analysis and acceptable internal consistency (Cronbach’s α ranging from 0.71 to 0.87 across subscales) ([13]). Its application in university samples has further supported its reliability and construct validity for assessing volitional components in relation to physical activity behavior, making it suitable for research and intervention contexts.

### 2.5. Psychological Tests

The following psychological tests were administered to all participants (*N* = 202) to assess psychological well-being.

#### 2.5.1. Perceived Stress Scale (PSS-10)

We used the Italian version of the Perceived Stress Scale ([4]), which is a 10-item self-report measure of perceived stress to evaluate the degree to which respondents appraise events as stressful during the past month. Items are rated on a 5-point Likert scale (from 0 = never. to 4 = very often), and higher total scores indicate greater perceived stress.

#### 2.5.2. World Health Organization—Five Well-Being Index (WHO-5)

This is a questionnaire that is commonly used to assess subjective well-being and mental health ([33]). It is a self-reported measure that consists of five items, each focusing on different aspects of well-being. Participants are asked to rate each item on a 6-point Likert scale ranging from 0 to 5, with higher scores indicating greater well-being. The items assess feelings of happiness, interest in daily activities, energy levels, and overall satisfaction with life.

#### 2.5.3. Short Form of the Psychological General Well-Being Index (PGWBI_S)

The questionnaire assesses both the individual’s subjective well-being and psychological health. The Italian version ([37]) is a short self-report measure that consists of six items, each one covering one dimension of well-being: anxiety, depressed mood, positive well-being, self-control, general health, and vitality. It is designed to capture both positive and negative aspects of well-being in the past month, providing a comprehensive assessment of an individual’s psychological state. Each item is rated on a 6-point Likert scale. ranging from 0 (none of the time) to 5 (all of the time).

Both the WHO-5 Well-Being Index and the General Well-Being Index—Short Form (GWBI-S) were included to capture complementary dimensions of psychological well-being. The WHO-5 primarily assesses positive affect and subjective vitality, reflecting hedonic aspects of well-being, whereas the GWBI-S provides a broader evaluation encompassing both positive and negative emotional states including stress and mood regulation. Using both instruments allowed for a more comprehensive and cross-validated assessment of the participants’ overall mental well-being.

### 2.6. Statistical Analysis

A post hoc power analysis was conducted for a medium effect size (Cohen’s *d* = 0.5) at α = 0.05, two tails. The analysis indicated that the study had high power to detect differences between medical and non-medical students (power (1-beta error prob) ≈ 0.84) and between male and female participants (power ≈ 0.93), indicating sufficient sensitivity to identify medium-sized effects. Descriptive statistics were first computed for all variables. Continuous variables are presented as means and standard deviations (sd).

Independent samples *t*-tests were performed to compare anthropometric variables, Mediterranean diet adherence, physical activity levels (expressed in METs), and the results of the psychological test between medical students and students from other faculties.

Moreover, an independent samples *t*-test was performed to compare anthropometric variables, Mediterranean diet adherence, physical activity levels (expressed in METs) and the results of psychological test between male and female students. In addition to statistical significance, effect sizes (Cohen’s, *d*) were computed to quantify the magnitude of these differences, following the interpretation guidelines proposed by [23] ([23]).

A two-way ANOVA was conducted to examine the effects of faculty and gender (the possible interaction faculty*gender) on anthropometric variables, Mediterranean diet adherence, physical activity levels (expressed in METs), and the results of psychological test. Considering that IPAQ determines three different categories of physical activity (low, moderate and high), a chi-square test of independence was conducted to analyze categorical associations, in particular, the relationship between gender and physical activity level (IPAQ categories by gender) and between faculties and physical activity level (IPAQ categories by faculty). Effect sizes were estimated using Cramer’s *V* and interpreted according to widely accepted benchmarks for small (*V* = 0.1), medium (*V* = 0.3), and large (*V* = 0.5) effects ([20]). Finally, Pearson’s and Spearman’s correlation analyses were performed to explore associations between physical activity (IPAQ in METs) and volitional factors measured by the VEQ-I (total and two subscale scores VEQ_VF and VEQ_IF) and psychological variables (PSS-10, WHO-5 and PWGB). Since *r* inherently represents an effect size, its magnitude was interpreted according to conventional benchmarks for small (≈0.10), medium (≈0.30), and large (≈0.50) effects ([23]). A principal component analysis (PCA) was performed to explore the structure of the dataset and reduce dimensionality prior to group comparison. PCA was applied as an exploratory data-reduction technique to identify latent dimensions underlying the multiple (anthropometric, behavioral, and psychological) variables included in the survey. The rationale was to simplify the dataset by grouping correlated measures into a smaller number of components that captured the main sources of variance.

All statistical analyses were performed with SPSS statistical package (Version 24.0 for Windows; SPSS Inc., Chicago, IL, USA). Statistical significance was set at *p* ≤ 0.05.

## 3. Results

### 3.1. Gender Differences

Independent-samples *t*-tests were conducted to examine the differences between male and female participants across anthropometric, behavioral, and psychological variables (Table 1). See Appendix A for the two subscale scores VEQ_VF and VEQ_IF.

Results showed that males had significantly higher weight and height compared with females (76.8 ± 12.1 kg vs. 60.8 ± 15.1 kg; 179.1 ± 6.1 cm vs. 164.9 ± 6.2 cm), with *t*(200) = –7.84 and –15.8, respectively, *p* < 0.001. These differences reflected large effect sizes (*d* = 1.18 and 2.24, respectively). Males also had significantly higher BMI values than females (*t*(200) = –2.3, *p* = 0.02), corresponding to a small-to-moderate effect (*d* = 0.32), as expected.

Psychologically, females reported significantly higher perceived stress (PSS-10: 19.7 ± 5.3 vs. 15.1 ± 5.5) *t*(200) = 5.84, *p* < 0.001, with a large effect size (*d* = 0.84). In contrast, males reported significantly higher psychological well-being scores on both the WHO-5 (*t*(200) = –2.83, *p* < 0.01, *d* = 0.40) and PGWBI-S (*t*(200) = –2.7, *p* = 0.007, *d* = 0.39), both indicating moderate effects.

In Figure 1, the psychological scores are shown for the male and female students.

#### Association Between Gender and Physical Activity Level (Chi-Square Analysis)

A chi-square test of independence was conducted to examine the association between gender and physical activity levels as classified by the IPAQ categories (low, moderate, high) (Table 2).

Results revealed a statistically significant association between gender and physical activity level (IPAQ categories) χ^2^(2, *N* = 202) = [8.26 χ^2^], *p* = 0.01, *V* = 0.20, indicating a small-to-moderate effect size. Specifically, females were more represented in the moderate category (29.6%) compared with males (15.6%), whereas males were slightly more represented in the high category (71.4%) than females (51.2%).

Overall, the majority of participants (58.9%) fell into the high physical activity category, but the distribution across categories significantly differed by gender.

### 3.2. Academic Programs Comparison

We conducted a comparison of questionnaire scores across different academic programs specifically, medicine and others to determine whether the perceived stress levels differed significantly by field of study.

Independent-samples *t*-tests were conducted to compare medical (*n* = 157) and non-medical students (*n* = 45) across anthropometric, behavioral, and psychological measures (Table 3).

There were no significant differences between groups in terms of weight, height, BMI, dietary habits, perceived stress (PSS-10), general well-being (PGWB-S), or volition scores (VEQ-I total).

Although non-medical students reported higher levels of physical activity compared with medical students (IPAQ METs: 7717.6 ± 12,198.9 vs. 4978.2 ± 9311.5, respectively), this difference did not reach statistical significance, *t*(200) = –1.61, *p* = 0.107.

A significant difference emerged for psychological well-being as measured by the WHO-5: non-medical students scored higher than medical students (15.62 ± 4.03 vs. 13.97 ± 4.91, respectively), *t*(200) = –2.05, *p* = 0.041, *d* = −0.35, indicating a small-to-moderate effect favoring non-medical students.

#### Association Between Faculty and Physical Activity Level (Chi Square Analysis)

A chi-square test of independence was performed to examine the relationship between faculty (medical vs. non-medical) and physical activity levels, as classified by the IPAQ categories (low, moderate, high) as shown in Figure 2.

Results revealed a significant association between faculty and IPAQ physical activity level, χ^2^(2, *N* = 202) = [16.08 χ^2^], *p* = 0.000, *V* = 0.28, indicating a moderate effect size. Notably, none of the non-medical students fell into the low activity category, while 21.7% of the medical students did. Furthermore, a higher proportion of non-medical students were classified as having high physical activity (82.2%) compared with medical students (52.2%).

These findings suggest that non-medical students reported higher levels of physical activity than their medical peers.

### 3.3. Interaction Between Faculty and Gender

A two-way ANOVA was conducted to examine the effects of faculty and gender on anthropometrics measures, diet score, IPAQ METs, VEQ-I total score, and psychological test scores. The interaction between faculty and gender was non-significant, indicating that the effect of gender on anthropometric measures, diet score, IPAQ-Mets, VEQ-I, and psychological test scores did not depend on faculty.

### 3.4. Correlations Between Anthropometric, Behavioral, and Psychological Variables

Pearson correlation analyses were conducted to examine associations among anthropometric measures, physical activity (IPAQ in METs), volition (VEQ-I), and psychological measures (Table 4). Since *r* inherently represents an effect size, the magnitude of the correlations was interpreted following conventional benchmarks for small (≈0.10), medium (≈0.30), and large (≈0.50) effects ([23]).

As shown in the highlighted yellow and orange cells in Table 4, several expected and well-established relationships were confirmed. For instance, weight and BMI demonstrated a strong positive correlation (*r* = 0.877, *p* < 0.01), which is consistent with their mathematical and physiological interdependence. Similarly, height and weight (*r* = 0.582, *p* < 0.01) and height and BMI (*r* = 0.582, *p* < 0.01) also showed moderate correlations, reflecting their role in anthropometric profiling.

A strong positive correlation was observed between WHO-5 and PGWBI-S (*r* = 0.665, *p* < 0.01), supporting the internal consistency and convergent validity of the two instruments in measuring psychological well-being in the sample.

Moreover, higher stress (PSS-10) was strongly associated with lower well-being (WHO-5: *r* = −0.540, *p* < 0.01; PGWBI-S: *r* = −0.644, *p* < 0.01).


*Anthropometric variables:*


Age was positively correlated with healthier dietary habits (*r* = 0.233, *p* < 0.01), but negatively associated with stress levels (PSS-10; *r* = −0.138, *p* < 0.05), and weight was negatively associated with VEQ-I (*r* = −0.220, *p* < 0.01) and perceived stress (*r* = −0.220, *p* < 0.01).

The more interesting results are shown in the green and blue squares.

Height was positively correlated with VEQ-I (*r* = 0.147, *p* < 0.05), WHO-5 (*r* = 0.157, *p* < 0.05), and PGWBI-S (*r* = 0.157, *p* < 0.05), but negatively with PSS-10 (*r* = −0.323, *p* < 0.01).


*Behavioral variables:*


IPAQ was positively associated with VEQ-I (*r* = 0.173, *p* < 0.05).

VEQ-I showed strong positive correlations with well-being (WHO-5: *r* = 0.333, *p* < 0.01; PGWB: *r* = 0.403, *p* < 0.01), while being strongly negatively correlated with PSS-10 (*r* = −0.401, *p* < 0.01). See Appendix A for the two subscale scores VEQ_VF and VEQ_IF.

Mediterranean diet adherence showed strong positive correlations with age (*r* = 0.233, *p* < 0.01) and well-being (WHO-5: *r* = 0.178, *p* < 0.05).

### 3.5. Principal Component Analysis (PCA)

PCA was conducted to reduce the dimensionality of the dataset and explore potential group differences between medical students (PC2_MED) and students from other faculties (PC2_Other). The percentage of variance explained was PC1: 27.9%, PC2: 18.1%, PC3: 11.6%, and PC4: 11.3%, respectively. The first two principal components (PC1 and PC2) were extracted and plotted to visualize the distribution of participants across the reduced dimensional space.

The loadings of each variable across the first four principal components (PC1–PC4) are reported in Table 5. Only factor loadings above |0.60| were considered meaningful for interpretation.

#### 3.5.1. PC1—Psychological Well-Being/Stress Axis

PC1 accounted for the largest proportion of explained variance and was characterized by strong and opposing loadings of PSS-10 (*r* = –0.834), PGWB-S (*r* = 0.807), WHO-5 (*r* = 0.771), and VEQ-I (*r* = 0.609). This component clearly represents a mental health continuum, contrasting high perceived stress (negative loading) with positive emotional well-being and self-regulation in eating behavior. It can be interpreted as a psychological health factor, where higher scores correspond to better emotional balance and lower stress.

#### 3.5.2. PC2—Morphological Dimension

PC2 was mainly associated with the anthropometric variables weight (*r* = 0.846) and height (*r* = 0.737). These loadings suggest that PC2 represents a morpho-physical dimension, independent from psychological traits.

#### 3.5.3. PC3—Age and Diet Factor

PC3 showed high positive loadings for age (*r* = 0.712) and diet quality (*r* = 0.703). This pattern indicates a possible age-related trend in dietary behavior, where older participants reported healthier eating habits. This can be interpreted as a lifestyle maturity axis.

#### 3.5.4. PC4—Physical Activity

PC4 was dominated by a strong loading from IPAQ (physical activity level) (*r* = 0.851). This component captures variance related specifically to habitual physical activity, with minimal cross-loading from other dimensions, supporting its interpretability as an independent behavioral factor.

This approach facilitated a more integrated interpretation of the data and enabled meaningful group comparisons (e.g., between medical and non-medical students) based on composite indicators rather than individual variables, thereby reducing redundancy and enhancing interpretability. In future, PCA can be used in status monitoring and intervention efficacy evaluation.

## 4. Discussion

The present study investigated the interrelations among the anthropometric, behavioral, and psychological variables, with a specific focus on the associations between physical activity, volitional determinants, and psychological well-being in a university student population. Additionally, differences across gender and academic discipline were taken into account. The findings confirmed significant gender-related differences across several domains, reflecting patterns already noted in the previous literature. As expected, males showed higher anthropometric values (weight, height, and BMI), which may partially reflect biological differences rather than lifestyle disparities. More interestingly, males reported lower perceived stress and higher well-being, suggesting a potentially greater resilience or more effective coping mechanisms during university life. Their higher scores in volitional facilitation and overall volition (VEQ-I) further indicate a stronger ability to regulate goal-directed behaviors such as maintaining regular physical activity. This pattern aligns with previous evidence ([32]) showing that male university students often report higher volitional self-confidence and better strategies for coping with failure, which may, in turn, foster greater engagement in physical activity. Conversely, the lower volitional and well-being scores among females could reflect greater exposure to academic and emotional stressors or differences in motivational processes, emphasizing the need for gender-sensitive health promotion strategies within university contexts.

Consistent with previous research ([7]; [16]), male students exhibited higher levels of physical activity (IPAQ score) along with lower perceived stress and greater psychological well-being compared with females. Moreover, in our study, there was a significant association between IPAQ categories and both genders. It is noteworthy that most students in this study reported high levels of physical activity. This result aligns with recent evidence showing that many university students remain physically active despite academic and social challenges. [3] ([3]) highlighted that supportive campus environments and peer influence play crucial roles in sustaining exercise habits during higher education. Similarly, [25] ([25]) found that even under restrictive conditions, such as the COVID-19 lockdowns, students demonstrated resilience in maintaining or re-establishing active routines. [21] ([21]) also reported that a considerable proportion of university students met the recommended activity levels, reinforcing the importance of this life stage for developing lasting healthy behaviors. Overall, these findings suggest that universities can serve as ideal settings for implementing evidence-based interventions to support physical activity and promote both physical and psychological well-being.

Regarding academic program differences, students from non-medical programs showed significantly higher psychological well-being (WHO-5), though no significant differences emerged in volitional scores or MET-based physical activity levels. However, the distribution of IPAQ categories revealed that non-medical students were significantly more likely to fall into the high physical activity group, with no participants in the low category, in contrast to medical students who were more evenly distributed across the three levels. This finding may reflect the greater academic pressure or time constraints experienced by medical students, potentially reducing their engagement in regular physical activity. Although non-medical students did not differ in volition scores, they scored higher on psychological well-being (WHO-5) and were more likely to fall into the IPAQ high activity category. This may reflect lower academic stress or different lifestyle demands compared with medical students—a hypothesis in line with general observations on academic workload impacting student health behaviors. However, a significant association between the IPAQ categories and faculty was confirmed, which highlights a great involvement in physical activity. This is a key point considering that recent evidence has highlighted the effectiveness of various physical activity interventions in managing stress and improving quality of life among university students. Moreover, moderate- to vigorous-intensity physical activities—such as running, yoga, and resistance training—were consistently associated with improvements in psychological functioning, particularly in reducing perceived stress and promoting mental health ([9]).

### Correlations

On the psychological side, a significant negative correlation emerged between perceived stress and indicators of well-being; these findings support the existing literature on the inverse relationship between stress and subjective well-being, confirming the validity of the constructs and the coherence of the scales used in this sample.

These associations, while not novel, are important as internal validation checkpoints. They reinforce the construct reliability of the variables and confirm the internal consistency of the questionnaire battery in a young adult university population.

Correlational analyses highlighted meaningful associations between volitional factors and psychological outcomes. Volitional facilitation (VEQ-VF) positively correlated with psychological well-being (PGWBI_S, WHO-5) and negatively with perceived stress, supporting its protective role. Conversely, volitional inhibition (VEQ-VI) was associated with higher stress and lower well-being, suggesting its relevance as a psychological barrier to goal achievement. Importantly, total physical activity (METs) was positively associated with VEQ-I scores and volitional facilitation, reinforcing the central role of motivation and self-regulation in sustaining exercise behaviors.

The strong associations found between volitional facilitation (VEQ-VF), physical activity, and psychological well-being support the theoretical framework of volition as a self-regulatory process crucial for health behaviors ([13]). Moreover, it is important to consider that VEQ-I was originally validated ([10]) as a reliable measure of self-regulation in exercise settings, with demonstrated predictive validity for exercise participation.

Our findings showed significant negative associations between volitional inhibition factors (VEQ-VI) and both perceived stress (PSS-10) and indicators of psychological well-being (WHO-5, PGWBI_S). This suggests that students experiencing greater volitional inhibition—such as procrastination, intrusive thoughts, or dependence on external approval—are more likely to report higher stress and lower well-being. These results are consistent with theoretical frameworks that emphasize the crucial role of inhibitory control in emotional regulation and adaptive coping. For instance, inhibition is considered a key component of self-regulatory capacity, essential for suppressing maladaptive impulses and maintaining goal-directed behavior under stress ([10]). Broadly, inhibition reflects deficits in cognitive control—a known risk factor for stress vulnerability ([40]). Although the present study did not explore physiological mechanisms, it is important to acknowledge that prolonged exposure to stress may have consequences beyond psychological well-being. Previous evidence has shown that stress can alter gut physiology and immune regulation ([22]) and may also influence gastrointestinal function, sleep quality, and overall health ([19]; [36]). Moreover, sustained stress exposure has been associated with detrimental cardiovascular effects, whereas regular physical activity may exert a protective role by reducing stress-induced risks ([29]). Future studies could therefore examine integrated interventions that combine psychological support, nutrition, and physical activity to mitigate the multifaceted impact of stress and promote long-term health among university students. Moreover, physical activity and eating habits may act as moderating factors that mitigate the negative impact of stress on the students’ perceived academic competence and study capacity, enhancing cognitive functioning and potentially buffering the detrimental effects of stress on academic performance ([38]).

This result is in line with another Italian study ([28]) that demonstrated the absence of significant differences in lifestyle–well-being associations between healthcare and non-healthcare students even if, in our study, non-medical students reported a better well-being outcome.

Finally, the PCA identified four distinct latent factors: (1) psychological well-being vs. stress, (2) morpho-physical structure, (3) age-related lifestyle behaviors, and (4) physical activity engagement. These findings confirm the multidimensional nature of student well-being, integrating emotional, behavioral, and physiological components.

The main limitations of the present study are as follows. (1) The sample size and representativeness: Although the total sample (*N* = 202) was adequate for detecting medium effects in between-group comparisons (Cohen’s *d* ≈ 0.5, power = 0.84, α = 0.05), it may still not be fully representative of the general university population. The unbalanced distribution between medical and non-medical students as well as between genders may have influenced some subgroup comparisons, thus limiting the generalizability of the findings. Future research should consider stratified or larger samples to better capture the variability of university student populations. (2) Cross-sectional design: Measurements were conducted at a single time point. Longitudinal assessments across different academic periods (e.g., class attendance, exam sessions) would allow for a more accurate evaluation of the temporal dynamics of stress, physical activity, and well-being. (3) Self-reported anthropometrics: Height and weight were self-reported, which may have introduced bias in the BMI calculation due to under- or overestimation. Objective measures are recommended in future studies to ensure greater accuracy. (4) Self-report bias in behavioral measures: Although validated instruments were used, all questionnaires relied on self-reported data, which are inherently prone to recall and social desirability biases. This is particularly relevant for dietary assessments, as food frequency questionnaires may not fully capture habitual intake. Meta-analytic evidence confirms variability in their reproducibility and accuracy across populations ([6]). Future research should consider complementing self-report measures with objective indicators (e.g., wearable sensors, dietary biomarkers) to enhance reliability and validity.

Future studies should consider the effects of multicomponent university physical activity programs on the psycho-physical well-being and their relationship with behavioral and stress variables among medical and non-medical university students. These programs could include onsite sport practice, active mobility to and inside the college campus, individualized sports and physical exercise counseling service, structured physical exercise classes, etc. Moreover, as suggested by several studies ([12]; [8]; [35]), mindfulness-based interventions and other beneficial techniques, such as yoga, could help medical students to cope with stress management, improving mental well-being and emotional regulation.

## 5. Conclusions

These findings emphasize the crucial role of volitional processes and psychological well-being in shaping university students’ health behaviors. Gender and faculty-related differences emerged, with males and non-medical students reporting higher physical activity levels and better well-being outcomes. Volitional facilitation appears to promote adaptive behaviors, whereas volitional inhibition is linked to increased stress.

From a practical perspective, universities should implement targeted programs to strengthen volitional skills through self-regulation and motivation workshops, alongside structured opportunities for regular physical activity. Medical students may particularly benefit from stress management interventions such as mindfulness or yoga-based programs. Integrating nutrition education and initiatives promoting adherence to the Mediterranean diet into campus health services could further enhance both physical and psychological health.

Additionally, peer mentoring, psychological counseling, and multidisciplinary well-being centers may help create supportive environments that encourage healthy lifestyles and emotional resilience. Future institutional policies should therefore adopt a holistic approach, integrating physical, psychological, and educational dimensions of student health.

## Figures and Tables

**Figure 1 behavsci-15-01468-f001:**
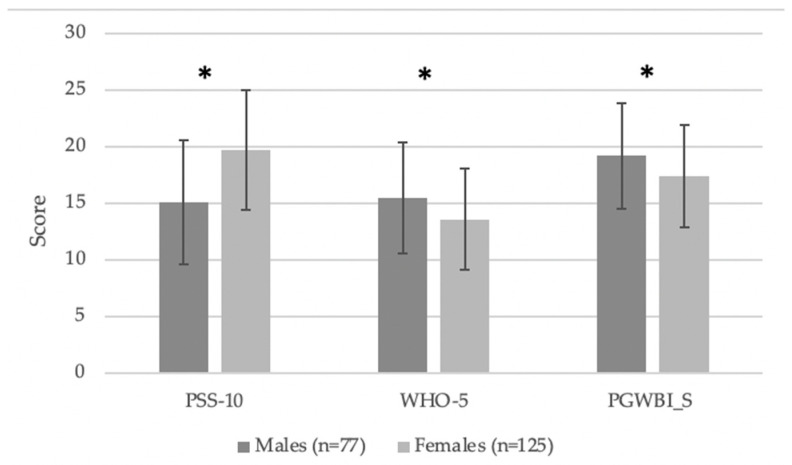
Male and female individual scores (mean ± sd) for the psychological test. * *p* < 0.001.

**Figure 2 behavsci-15-01468-f002:**
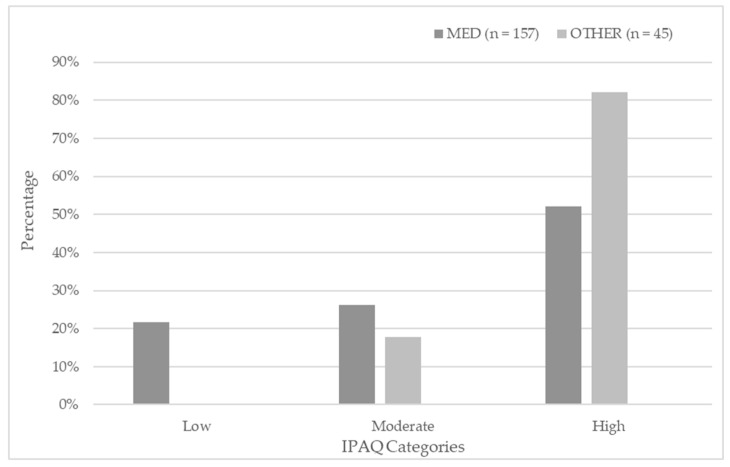
Distribution of physical activity levels (IPAQ categories) among medical and non-medical students. The chart shows the percentage of participants classified as low, moderate, or high physically active within each group.

**Table 1 behavsci-15-01468-t001:** Results of the independent-samples *t*-tests comparing male vs. female participants across the following variables: anthropometric measures, diet score, IPAQ MET-equivalents (METs), VEQ-I total score, and psychological test scores. Means, standard deviations (m ± sd), t-values, effect size (*d*), and two-tailed *p*-values are shown (in bold *p* < 0.05).

Variable	All (*n* = 202)	Males (*n* = 77) (m ± sd)	Females (*n* = 125) (m ± sd)	*t*	*d*	*p*
Weight (kg)	66.93 ± 16.09	76.8 ± 12.1	60.8 ± 15.1	−7.84	1.18	**0.00**
Height (cm)	170.35 ± 9.33	179.1 ± 6.1	164.9 ± 6.2	−15.8	2.24	**0.00**
BMI (kg/m^2^)	22.93 ± 4.58	23.8 ± 3.1	22.3 ± 5.2	−2.3	0.32	**0.02**
Diet	7.15 ± 2.07	7.2 ± 2.3	7.1 ± 1.9	−0.33	0.05	0.73
IPAQ (METs)	5588.5 ± 10,058.6	6146 ± 5876	5245 ± 11,939	−0.61	0.09	0.53
VEQ-I	0.76 ± 0.85	1.04 ± 0.80	0.58 ± 0.84	−3.84	0.55	**0.00**
PSS-10	17.99 ± 5.86	15.1 ± 5.5	19.7 ± 5.3	5.84	0.84	**0.00**
WHO-5	14.34 ± 4.77	15.5 ± 4.9	13.6 ± 4.5	−2.83	0.40	**0.00**
PGWBI_S	18.09 ± 4.71	19.2 ± 4.7	17.4 ± 4.5	−2.7	0.39	**0.00**

**Table 2 behavsci-15-01468-t002:** Cross-tabulation of the IPAQ physical activity levels by gender.

IPAQ Category	Males (*n* = 77)	Females (*n* = 125)	Total (*N* = 202)
Low	10 (13.0%)	24 (19.2%)	34 (16.8%)
Moderate	12 (15.6%)	37 (29.6%)	49 (24.3%)
High	55 (71.4%)	64 (51.2%)	119 (58.9%)

**Table 3 behavsci-15-01468-t003:** Independent-samples *t*-tests comparing medical (MED) and non-medical (Other) students. Means, standard deviations (m ± sd), t-values, effect size (*d*), and two-tailed *p*-values are shown (in bold *p* < 0.05).

Type	Variable	MED (*n* = 157) (m ± sd)			Other (*n* = 45) (m ± sd)			*t*	*d*	*p*
Anthropometric	Weight (kg)	67.33	±	16.96	65.53	±	12.67	0.66	0.11	0.51
	Height (cm)	170.03	±	9.08	171.49	±	10.17	−0.92	−0.16	0.355
	BMI (kg/m^2^)	23.15	±	4.94	22.13	±	2.87	1.32	0.22	0.187
Behavioral	Diet	7.11	±	2.06	7.31	±	2.11	−0.55	0.09	0.577
	IPAQ (METs)	4978.2	±	9311.5	7717.6	±	12,198.9	−1.61	−0.27	0.107
	VEQ-I	0.72	±	0.89	0.88	±	0.71	−1.13	−0.19	0.257
Psychological	PSS-10	17.97	±	5.96	18.04	±	5.54	−0.07	−0.01	0.944
	WHO-5	13.97	±	4.91	15.62	±	4.03	−2.05	−0.35	**0.041**
	PGWB-S	17.9	±	4.59	18.73	±	5.11	−1.03	−0.18	0.303

**Table 4 behavsci-15-01468-t004:** Pearson correlation (*r*) among variables.

Variables	Age	Weight	Height	BMI	Diet	IPAQ	VEQ-I	PSS-10	WHO-5	PGWBI-S
**Age**	1				0.233 **			−0.138 *		
**Weight**		1	0.582 **	0.877 **				−0.220 **		
**Height**		0.582 **	1				0.147 *	−0.323 **	0.177 **	0.157 **
**BMI**		0.877 **		1						
**Diet**	0.233 **				1				0.178 *	
**IPAQ**						1	0.173 *			
**VEQ-I**			0.147 *			0.173 *	1	−0.401 **	0.333 **	0.403 **
**PSS-10**	−0.138 *	−0.220 **	−0.323 **				−0.401 **	1	−0.540 **	−0.644 **
**WHO-5**			0.177 *		0.178 *		0.333 **	−0.540 **	1	0.665 **
**PGWB-S**			0.157 *				0.403 **	−0.644 **	0.665 **	1

** *p* = 0.01; * *p* = 0.05.

**Table 5 behavsci-15-01468-t005:** Factor loadings.

Variables	PC1 (Psycho.)	PC2 (Anthrop.)	PC3 (Diet)	PC4 (Phys. Act.)
**PSS-10**	−0.834	0.010	0.054	0.127
**PGWB-S**	0.807	−0.295	−0.117	−0.228
**WHO-5**	0.771	−0.289	0.019	−0.164
**VEQ-I**	0.609	−0.135	−0.193	0.185
**Weight**	0.309	0.846	−0.021	0.030
**Height**	0.472	0.737	−0.019	0.066
**Age**	0.225	−0.010	0.712	0.234
**Diet**	0.238	−0.112	0.703	0.242
**IPAQ**	0.180	−0.159	−0.388	0.851

Psycho.: psychological variables; Anthrop.: anthropometric variables; Phys. Act.: physical activity.

## Data Availability

The raw data supporting the conclusions of this article will be made available by the authors on request.

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
