# Peer review of "Lifestyle, Volition, and Well-Being Among Medical and Non-Medical University Students: A Preliminary Study"

_behavsci, 2025, doi:10.3390/bs15111468_

Round 1

Reviewer 1 Report

Comments and Suggestions for Authors

Lines 50-62: The content is duplicative and should be summarized more concisely.

Lines 63-71 + 78-80: In my opinion, this is not suitable. Fortunately, universities place great emphasis on students acquiring practical skills and competencies in addition to theoretical knowledge. I would therefore ask the authors to phrase this in a much more sensitive manner.

At the outset, a distinction is made between health sciences students and non-health sciences students (L48). As the paper progresses, it becomes clear that what is actually meant is a distinction between medical students and non-medical students (L84). This is highly problematic, as sports students are counted among the non-medical students, and one focus of the paper is precisely in the area of physical activity. This requires a detailed explanation by the authors. Shouldn't medical students necessarily be counted along with sports students? Doesn't the sports study program have a very significant health-related aspect? The sample size should not be used as an argument here. What other study programs are included in the “other” category? Does the category “medicine” also include dentistry? Background information on the study programs (content, focus areas, etc.) is required. This circumstance also has an impact on the discussion of these results (e.g. Line 351-355).

How many of the “other” students (n=45) are sport science students?

There are a number of simple errors in the manuscript (L167, L298, L418, L442). The authors should read through the text again critically.

Table 2: Error in the table: 46.2% should be 71.4%.; The percentage in line 219 is also incorrect.

Testing for differences between the sexes should not be carried out for its own sake, but should be guided by theory and empirical evidence. There are no comments on this in the introduction. Since the topic occupies a large part of the results and discussion, it should also be taken into account in the objectives.

When naming the categories (physical activity: low, moderate, high), quotation marks are sometimes used and sometimes not. Please (don't) use them consistently.

What exactly was the idea behind including both WHO-5  and GWBI_S in the survey? What are the differences and similarities between the constructs?

Diet is sometimes capitalized, sometimes written in lowercase.

Table 6: Authors should describe the PCs (e.g. PC 1 to PC1 – Mental well-being/stress axis) in detail here.

Is there an entrance test for sports studies in Italy? This may explain differences in weight an BMI.

Line344-348: This should be backed up with current literature.

Line404-411: These remarks seem out of place. What exactly is the point behind them in the context of the question?

Reviewer 2 Report

Comments and Suggestions for Authors

The manuscript addresses a highly relevant public health issue, namely lifestyle habits among university students, with a particular focus on differences between medical and non-medical students. The topic is timely and of interest to both researchers and practitioners in health psychology and education. The study is well structured, uses validated instruments, and employs a broad range of statistical analyses. The discussion connects the findings to recent literature and the limitations are openly acknowledged. However, some aspects require clarification and revision to improve the clarity, consistency, and impact of the work. Below I provide specific comments and suggestions.

Title and Abstract

Lines 1–3: The title “Healthy Lifestyle Habits in University Students: a preliminary study” is too generic. Suggestion: make it more specific, e.g., “Lifestyle, Volition and Well-Being among Medical and Non-Medical University Students: A Preliminary Study”.

Lines 19–37 (Abstract): The abstract is clear but overly technical. For example, “volitional facilitation (VEQ-VF)” may not be easily understood by a broad readership. Suggest simplifying and adding one or two sentences on practical implications (e.g., how universities can support students’ health).

Introduction

Lines 41–47: The sentence “Research shows health patterns emerge during these years, including poorer eating, less exercise, worse sleep, and increased stress” repeats information presented later. Condense to avoid redundancy.

Lines 52–60: The research question (“do students learning to care for others’ health effectively apply these principles…”) is repeated in slightly different wording. Streamline into one clear statement.

Lines 83–87: The phrase “being a preliminary study, the need to know what are the habits…” is unclear. Suggested rephrasing: “As a preliminary study, it is important to investigate lifestyle habits of students at the beginning of their academic journey.”

Methods

Lines 89–93: Recruitment procedure is not specified. Was it convenience sampling? Voluntary participation? Please clarify.

Lines 99–101: Height and weight were self-reported. This limitation should be explicitly highlighted in the Discussion as it may bias BMI.

Lines 168–187 (Statistical Analysis): The use of PCA is not sufficiently justified. Please clarify the rationale (exploratory structure? reducing variables for group comparison?).

Results

Lines 200–209 (Table 1): The text repeats numbers already in the table. Suggest focusing only on significant differences.

Lines 216–219: Inconsistency. The text states “males were slightly more represented in the High category (46.2%) than females (51.2%)”, but the values indicate the opposite (females higher). This must be corrected.

Lines 239–241: The difference in WHO-5 (p = 0.041) is statistically significant, but likely with a small effect size. Reporting effect sizes would improve interpretation.

Discussion

Lines 326–339: The discussion restates results (e.g., males showing lower stress and higher well-being) rather than interpreting them. Suggest shortening descriptive repetition and emphasizing interpretation.

Lines 408–417: The section on microbiota, probiotics, and gastrointestinal issues is interesting but outside the scope of this study. Suggest removing or framing as “future research directions”.

Lines 429–433: PCA interpretation is clear (psychological well-being/stress, morphological, diet/age, physical activity), but the relevance to the research question should be better explained. How do these components advance understanding of differences between medical and non-medical students?

Conclusion

Lines 449–456: Conclusions are too generic. Recommendations should be more concrete, such as:

  • Universities should implement programs to strengthen volitional skills (e.g., self-regulation workshops).
  • Structured physical activity opportunities should be made widely available.
  • Medical students in particular may benefit from stress management interventions (e.g., mindfulness, yoga).
  • Promoting adherence to the Mediterranean diet could be integrated into campus health programs.

Reviewer 3 Report

Comments and Suggestions for Authors

Thank you for the opportunity to review this timely and relevant manuscript. The lifestyle habits and health behaviors of the future medical workforce are undoubtedly an important topic, and the dataset and analyses presented here are of merit. However, I have several concerns that need to be addressed before I can recommend this manuscript for publication.

Introduction

Line 58 ff. The latter part of the introduction is notably weaker than the first half. References disappear altogether, and more casual language is introduced (e.g., “simply teaching people facts isn’t enough to change behaviors”). Please revise this section to ensure consistency with standards of academic writing and provide appropriate referencing.

Methods

Line 89. “ys” is not a standard abbreviation; please use years or yrs.

Line 89 ff. Means and standard deviations are presented here before the concepts are introduced in the statistics section. I recommend restricting this part to details on recruitment and presenting quantitative cohort parameters at the beginning of the Results section.

Line 93. Other students are not described in the same detail (only N is provided, without age or gender distribution). This should be clarified for comparability.

Please specify whether gender was self-reported.

Line 100. All abbreviations must be introduced at first mention; e.g., define BMI.

Line 102 ff. The MEDAS instrument requires a more detailed description, including its validation with appropriate references. The time frame for the nutrition assessment must also be clarified—was it the last 48 hours, two weeks, or another period? How can this measure be extrapolated to reflect habitual diet and long-term stability?

The same clarification and validation are needed for IPAQ and VEQ/I. Please discuss their reliability and validity.

Line 168 ff. Statistical analysis. This section should begin with descriptive statistics (means and SDs or medians and ranges).

Line 175 ff. Consider reporting an effect size appropriate for ANOVA, such as generalized eta squared.

A power calculation is missing. On what basis was the sample size determined? Even if not used in planning, please provide at least a post-hoc power analysis at α = 0.05.

Line 186. The rationale for using principal component analysis is not sufficiently explained and needs elaboration.

Results

Line 190 ff. Begin with descriptive statistics. Include a Table 1 describing the entire study population, not limited to medical students.

Line 268 ff. The use of bullet points is highly unusual in academic writing and gives an unprofessional impression. Please reformat.

Line 268. While numerous tables and p-values are reported, effect sizes are missing. Reporting effect sizes would provide important information about the magnitude of the observed associations.

The results section is dense with statistical parameters and difficult to follow. Please consider adding one or two figures visualizing key findings (e.g., adherence to the Mediterranean diet and WHO-5 scores).

Discussion

Line 434. The limitations section is too brief and vague. For example, the statement that the sample size may be insufficient is speculative without a sample size or power calculation. Please provide evidence to support or refute this.

A critical discussion of the reliability of the tools used is needed. In particular, self-reported dietary questionnaires are known to be prone to bias. This limitation should be explicitly acknowledged and discussed with reference to the relevant literature.

Overall recommendation: The study addresses an important and timely topic with potential relevance for the field. However, the manuscript requires substantial revisions in methodological rigor, clarity of reporting, and adherence to academic standards before it can be considered for publication.

Round 2

Reviewer 1 Report

Comments and Suggestions for Authors

The manuscript now reads much more clearly.

Reviewer 2 Report

Comments and Suggestions for Authors

After reviewing the revised version of the manuscript, I acknowledge that the authors have consistently and satisfactorily addressed the previously raised concerns. The title was aligned with the study objectives, the methodological justification for the PCA was clarified, numerical inconsistencies were corrected, and the Discussion section was improved by reducing descriptive repetition and strengthening interpretative depth. Methodological limitations, including the self-reported anthropometric data, were appropriately acknowledged, and the practical implications were made more concrete. Overall, the revisions have enhanced the clarity, analytical rigor and practical relevance of the manuscript. I have no further comments at this stage.

Reviewer 3 Report

Comments and Suggestions for Authors

Thank you for revising the manuscript. I'm convinced the edits have further strengthened the article's methodological rigor and relevance.

I recommend publishing it in the present form.